# Effect of Different Levels of Extruded Coffee (*Coffea arabica*) Pulp Flour on the Productive Performance and Intestinal Morphometry of Cobb 500 Broiler Chickens

**DOI:** 10.3390/ani14081170

**Published:** 2024-04-13

**Authors:** Steven Antúnez, Nadia Fuentes, Marco Gutierrez, Fernando Carcelén, Fritz Trillo, Sofía López, Sandra Bezada, Virginia Rivadeneira, Samuel Pizarro, Jimny Nuñez

**Affiliations:** 1Laboratorio de Producción Avícola y Especies Menores, Facultad de Medicina Veterinaria, Universidad Nacional Mayor de San Marcos, Av. Circunvalación 28, San Borja 15021, Lima, Peru; steven.antunez@unmsm.edu.pe (S.A.); vrivadeneira@unmsm.edu.pe (V.R.); 2Instituto Veterinario de Investigaciones Tropicales y de Altura (IVITA), Facultad de Medicina Veterinaria, Universidad Nacional Mayor de San Marcos, Carretera Huaral-Chancay km 6.5, Huaral 15200, Lima, Peru; nfuentesn@unmsm.edu.pe; 3AG-RESEARCH S.A.C., Av. Alfonso Ugarte SN Sapallanga, Huancayo 12400, Junín, Peru; marco.gutierrez.tang@gmail.com (M.G.); sam20048130@gmail.com (S.P.); 4Laboratorio de Bioquímica, Nutrición y Alimentación Animal, Facultad de Medicina Veterinaria, Universidad Nacional Mayor de San Marcos, Av. Circunvalación 28, San Borja 15021, Lima, Peru; fcarcelenc@unmsm.edu.pe (F.C.); slopezg@unmsm.edu.pe (S.L.); sbezadaq@unmsm.edu.pe (S.B.); 5Departamento Académico de Producción Animal, Facultad de Zootecnia, Universidad Nacional Agraria La Molina, Av. La Molina s/n, Lima 15024, Lima, Peru

**Keywords:** coffee pulp, small intestine, animal performance, intestinal villus, dietary fiber

## Abstract

**Simple Summary:**

Coffee pulp is a byproduct of the coffee industry and represents an environmental problem; therefore, alternatives are being sought regarding its use in other industries, such as for animal feed. The study evaluated the feasibility of replacing part of the wheat bran provided in broiler chickens’ diets during the finishing stage with extruded coffee pulp meal. The results indicated that coffee pulp can replace up to 25% of wheat bran in chicken feed in that phase of production, which could offer a viable alternative, in addition to contributing to a more sustainable and environmentally friendly production process.

**Abstract:**

Coffee pulp is a by-product of the coffee industry. Due to conventional management techniques, it represents a severe environmental problem due to its negative impact on the soil (anaerobic fermentation and pH changes), water sources (the infiltration of pollutants into streams, acidification of water sources, and modification of microorganisms), and biodiversity (soil microbiology, fish, crustaceans, and other vertebrates). Therefore, it is essential to develop protocols for the treatment of this waste so that it can be used again in other productive activities under the circular economy approach. This means that all the waste from a production process can be reused, can generate value for the benefit of the producer, and, in turn, mitigate the environmental impact. The objective of this study was to evaluate the replacement of 5 levels of wheat bran (WB) with extruded coffee pulp flour (ECPF) as an alternative to a conventional fiber source in broiler finisher diets. A total of 300 Cobb 500 chickens in the finishing phase were assessed in the study, grouped in 5 treatments: T1, a conventional diet or control treatment (100% WB and 0% ECPF), T2 (75% WB and 25% ECPF), T3 (50% WB and 50% ECPF), T4 (25% WB and 75% ECPF), and T5 (0% WB and 100% ECPF). Feed intake, weight gain, feed conversion ratio (FCR), and intestinal morphometry (villus length: VL, villus width: VW, crypt depth: CD, villus height/crypt depth ratio: V/C, and villus surface area: VSA) were evaluated at the level of the duodenum, jejunum, and ileum. Feed intake decreased correspondingly as the ECPF in the diet was increased, with statistical differences (*p* < 0.01) between their averages; the most significant weight gain (834.61 g) was evidenced with the T2 treatment, this being statistically different (*p* < 0.01) from T4 and T5; similarly, the best FCR (1.58) was evidenced with the T2 treatment, followed by the control treatment T1 (with 1.64); however, they were not statistically different (*p* > 0.05). All treatment results were similar to the VL control samples in the three intestinal portions, except for the T5 in the jejunum, which showed statistical differences from the control. In VW, the treatment results were similar to the control samples of the jejunum and ileum; however, in the duodenum, the T5 results showed the highest value (172.18 μm), being statistically different (*p* < 0.05) from the other treatments being evaluated. For CD, it was only in the duodenum that the T2 and T3 treatments were similar to the control. Likewise, for V/C in the duodenum, only the T2 results were similar to the control. There was no significant difference in the VSA among the different treatment groups. T2 showed better production parameters without altering the intestinal villi. In conclusion, ECPF is a potential input for use to replace up to 25% of WB in the feed of broilers in the finishing phase.

## 1. Introduction

Coffee (*Coffea arabica* L.) is the most preferred beverage in the world. It is also the most important cash crop and the second most valuable international commodity after oil [1]. Peru is the ninth largest coffee exporter in the world, with 425,416 hectares dedicated to coffee cultivation, which represents 6% of the agricultural area and 25% of the Amazon area, with coffee plantations installed in 17 regions, 67 provinces, and 338 districts, generating employment for 223,738 Peruvian families [2].

Worldwide, more than 20 million tons of liquid (mead) and solid (pulp) waste are produced annually as by-products of coffee processing [3,4] which represents 56% of the fresh fruit weight [5], while the pulp constitutes between 40 and 50% of the weight of the fruit [6,7]. These wastes constitute a severe environmental problem due to their negative impact on the soil (anaerobic fermentation and pH changes), water sources (the infiltration of pollutants into streams, acidification of water sources, and modification of microorganisms), and biodiversity (soil microbiology, fish, crustaceans, and other vertebrates). Therefore, it is essential to develop more efficient and sustainable systems focused on the circular economy [8], such as producing feed for production animals.

The coffee pulp includes the skin, epidermis, exocarp, and mesocarp of the coffee cherry tree [7]; its structure contains caffeine, polyphenols, and tannins, which are a potential source of antioxidants [9,10]. Caffeine accounts for 1 to 2% of the dry weight of coffee [11], while the total phenolic compounds and the equivalent chlorogenic acid are concentrated to between 2.12 and 2.37 g/100 g of the dry weight [12,13,14]. Coffee pulp is a renewable resource that is rich in carbohydrates and has a minimal fraction of proteins (9.2%), lipids (2.6%), and a high content of dietary fiber (51.2%), highlighting the insoluble fraction [15]. The authors of [12] reported a high content of dietary fiber in coffee pulp (46.12%), which includes soluble and insoluble fiber with values of 9.13 and 36.99%, respectively. The latter is important because it promotes intestinal regulation; coffee by-products are considered to have a high potential for use in functional foods that are rich in fiber. They can also be metabolized by probiotic bacteria (Lactobacillus and Bifidobacterium) by producing beneficial metabolites (short-chain fatty acids).

The high concentrations of caffeine, polyphenols, and tannins act as antinutrients [16]; tannins have the ability to bind to the proteins or act as enzyme inhibitors, interfering with the biological availability of protein [17]. These antinutrients reduce the use of pulp in animal feed [18,19]. However, due to its nutritional characteristics, coffee pulp has been used in the feed of various production species, such as sheep [20,21,22,23,24,25] fighting cocks [24,26], guinea pigs (*Cavia porcellus* L.) [27], pigs [17], cattle [28,29,30,31], and fish [32,33,34].

Extrusion is a process in which the feed is subjected to thermal and mechanical energy for short periods, whereby high pressure and temperature generate substantial changes to the shape, structure, and composition of the feed [35] at low cost [36]. The technique is widely applied in the production of food for humans [37,38], aquatic animals, and pets [39] because it substantially improves the nutritional quality, protein digestibility, and antioxidant activity, and significantly reduces the antinutrients [40]. The extrusion of fibrous by-products favors the solubilization of dietary fiber, the release of phenolic compounds, and the modification of the functional and antioxidant properties of foods [41,42].

Chicken production is Peru’s most important livestock activity, accounting for 2% of the country’s gross domestic product. During the year 2022, the per capita consumption of chicken meat reached 47.33 kg/inhabitant/year [43], with chicken being the most widely consumed meat in the country [44]. This industry contributed 65% of the value to the Peruvian economy of livestock production, 24% of the value of agricultural production, and approximately 2% of the national gross domestic product (GDP) [45].

Integration is a strategy used by the poultry industry to reduce production costs, as it provides significant economic improvements to farms due to the elimination of intermediaries, the production of large volumes, and the reduction of transaction costs, thereby making the producers more competitive [46,47,48,49]. In Peru, 55% of poultry producers are integrated to reduce production costs (feed represents between 60 and 75% of the total production costs). Therefore, there is a constant search for inexpensive local alternative products to replace traditional inputs. Consequently, the aim of the present study was to evaluate extruded coffee pulp flour (ECPF) as a replacement for wheat bran (WB) in broiler feed.

## 2. Methodology

This research followed the following methodological framework: first, fresh coffee pulp (from a coffee farm) was obtained. Subsequently, the optimal extrusion process was carried out at the Laboratorio de Producción Avícola y Especies Menores (LPAYEM). Later, at the Laboratorio de Bioquímica, Nutrición y Alimentación Animal (LBNAA), various dietary formulations were developed, based on a nutritional analysis of the input. Finally, biological tests were conducted at the Instituto de Investigaciones Tropicales y de Altura (IVITA), located in Huaral, Lima (Figure 1).

### 2.1. Area of Study

The fresh coffee pulp was obtained from a coffee farm that was representative of those in the central rainforest of Peru (10°43′29″ S 75°04′43″ W), located in the District of Perené, Province of Chanchamayo. The coffee pulp comes from the coffee varieties known as Caturra and Típica, which have existed for eight years.

The cherry coffee tree was harvested manually, and the harvest was then transported to the milling area, where it was dry-pulped using a drum pulper (brand: Bonanza; engine: Honda 4.0 HP; approximate yield: 250 kg/h). The coffee pulp was pre-dried in situ at up to 25% humidity using solar dryers to ensure its preservation. It was then packaged in a transparent polyethylene bag and placed inside another black polypropylene bag, then it was labeled, sent first to the LPAYEM, then sent later to the LBNAA of the Faculty of Veterinary Medicine of the Universidad Nacional Mayor de San Marcos (Figure 1).

### 2.2. Extruded Coffee Pulp Flour (ECPF)

At the LPAYEM, the coffee pulp was dehydrated to 15% humidity using a rotary dryer (Galix Tech, Huancayo, Perú), and then it was passed through a hammer mill (Galix Tech) to obtain coffee pulp flour. It was then hydrated to 45% humidity and extruded in a twin-screw extruder (Galix Tech^®^) with a temperature of between 110 and 131 °C and 45% humidity. Finally, the extruded coffee pulp was cooled to the environmental temperature and then milled to obtain extruded coffee pulp flour. The presentations of the various forms of coffee pulp are presented in Figure 2.

### 2.3. Diets

For the formulation of food diets (Table 1), protein at 14.24% (AOAC 2011.11) [50], crude fiber at 9.83% (AOAC 962.09) [51], and metabolizable energy at 11791.60 kJ/kg [52] of ECPF were used as a basis. In addition, ECPF has phenolic antioxidants (at less than 1.00 mg/kg with the HPLC method) and caffeine (around 1.1 g/100 g, according to ISO 20481.2008 [53]). These values were introduced into the Dapp Nutrition^®^2.0 Formulation Software to balance the diet, added according to the levels of replacement of WB by ECPF for each treatment (T1, T2, T3, T4, and T5) and considering the nutritional requirements of the Cobb 500 birds of the 2022 line. 

The nutritional value of the diets used is shown in Table 2.

### 2.4. Biological Testing on Chickens

The evaluation of the Cobb 500 chickens was carried out in the sheds of the Poultry Demonstration Module of the IVITA of the FMV—UNMSM, located in the Lima Region (12°02′36″ S 77°01′42″ W), at an average temperature of 18.9 °C and with an annual rainfall of approximately 203 mm. An open type of shed with a gabled roof and a polished cement floor with a 5° inclination was used. The experimental units were made of plastic mesh and 3/4″ PVC pipe of 1 × 1.2 m long by 0.5 m high; the bed was made of rice husks at a thickness of 3 cm (Figure 1). 

A total of 300 male broilers in the finishing phase (35 to 41 days of age) were used, with the diets being distributed as 5 treatments involving 6 replicates of 10 chickens each. The conventional finishing phase diet included 6% WB, which was replaced according to the treatments being evaluated, where T1 is the conventional diet (WB: 100% and ECPF: 0%), along with T2 (WB: 75% and ECPF: 25%), T3 (WB: 50% and ECPF: 50%), T4 (WB: 25% and ECPF: 75%), and T5 (WB: 0% and ECPF: 100%). The measured amount of feed was supplied in a hopper trough according to the Cobb 500 table for 2022, and drinking water was supplied ad libitum via an automatic drinker. 

#### Variables Assessed

Feed intake was determined daily, using the difference between the feed offered and the residual feed. Weight gain was calculated using the difference between the final weight and the weight at the start of the experiment. The feed conversion ratio (FCR) was obtained via the quotient between the feed consumed and weight gain. Measurements were made using a digital scale (brand: JBC; capacity: 5: kg; accuracy: 0.1 g). 

At the end of the study, 6 chickens were randomly slaughtered per treatment (after 6 h of fasting), then 1-centimeter segments were extracted from the middle portion of the duodenum, jejunum, and ileum. The segments were fixed in 10% buffered formalin [54] and sent to a private laboratory, Crishistotecnologa, where a histological section and subsequent tissue-staining with hematoxylin–eosin were performed [55]. Finally, the measurements of villus length (VL), villus width (VW), Lieberkühn crypt depth (CD), and the villus height/crypt depth ratio (V/C) were performed following the methodology of Vallejos et al. (2015) [56]. Finally, the villus surface area (VSA) was determined using the following VSA equation, where VSA = 3.1416 × VW × VL/100 × 100 [57]. 

### 2.5. Statistical Analysis

The study considered a completely randomized design (after checking normality via the Shapiro–Wilk test and homoscedasticity via the Barlett test) for both the productive and intestinal tissue variables. The post hoc Duncan test at 5% significance was used to compare the means. The statistical package R (Version 4.02) was used, employing the *agricolae* and *lm* libraries.

## 3. Results

### 3.1. Feed Intake

The feed intake was similar to that of the control when ECPF replaced 25 and 50% of WB; however, when ECPF replaced 75 and 100% of WB, consumption decreased significantly, being statistically (*p* < 0.05) different from the control. On the other hand, the lowest consumption was evidenced with the treatment using 100% replacement of WB with ECPF (Table 3). 

Weight gain was greater (834.61 ± 34.86 g) when 25% of the WB was replaced by ECPF, with the birds showing a lower weight gain as the replacement dose was increased; however, this treatment (25% replacement) exceeded 24.81 g in weight gain per bird, compared to treatment 1 (100% of the WB) of the poultry diets (Table 3). On the other hand, treatment with 100% replacement of WB by ECPF showed a lower weight gain (662.15 ± 69.43 g) compared to the other treatments being evaluated, this being statistically different (*p* < 0.05) even from the control (Table 3). 

The FCR showed its lowest value (1.58 ± 0.07) for the treatment with a 25% replacement of WB by ECPF, being statistically similar (*p* > 0.05) to treatments 1, 3, and 4 (Table 3).

### 3.2. Morphometric Measurements

The VL in the duodenum and ileum did not show statistical differences (*p* > 0.05) between the means; however, at the level of the jejunum, the highest VL (1415.64 μm) was evidenced with treatment 3, with 50% replacement of the WB by ECPF, this being statistically different (*p* < 0.05) only when 100% of the WB was replaced (Table 4).

The VW in the duodenum showed statistical differences (*p* < 0.05) when 100% of ECPF was used, compared to the control. However, in the jejunum and ileum portions, there were no statistical differences (*p* > 0.05) between the various means (Table 4). 

The CD showed statistical differences (*p* < 0.05) in the duodenum, jejunum, and ileum, showing a lower crypt depth with the control treatment (0% ECPF), followed by the treatment with 25% replacement of wheat bran with ECPF (Table 4).

The relationship between villus length and crypt depth (V/C) was higher in the non-ECPF treatment in the three intestinal portions evaluated, this being statistically different (*p* < 0.05) from the other treatments at the jejunum and ileum level; however, at the duodenum level, there were no statistical differences between the samples for T1 (9.41 μm) and T2 (8.17 μm). 

The villus surface area did not show statistical differences among the three evaluated intestinal sections (Table 4).

## 4. Discussion

The replacement of 25% of wheat bran (WB) with extruded coffee pulp flour (ECPF), corresponding to 1.5% of the diet in the finishing phase of Cobb 500 broilers, showed similar results compared to the control, with the replacement not affecting the productive performance of the bird. Donkoh et al. (1988) [24] reported similar outcomes when using 2.5% of dry coffee pulp (*Coffea arabica*) in the diet of 8-week-old AF Bosbek line chickens, which did not affect food consumption, weight gain, and the feed conversion ratio; however, higher values significantly affected the results.

In this research, values higher than 1.5% of the diet decreased food consumption, due to the increase in tannins, caffeine, and free polyphenols (chlorogenic acid and caffeic acid), which influence the odor and flavor of the food [17]. Caffeine, a methylxanthine with bitter characteristics [58], affected consumption and weight gain, as demonstrated by the authors of [59]. Caffeine reduces weight gain due to lipolysis and increased motor activity, leading to energy wastage, and also increases animal thirst, enhancing nitrogen excretion through urinary evacuation [60].

The digestive tract physiology of modern chickens has undergone changes due to genetic improvements aimed at faster growth and better feed efficiency. This rapid growth demands a significant intake of highly digestible foods [61]. Their absorptive capacity depends on intestinal integrity, which includes the evaluation of villi and Lieberkühn’s crypts [62] and is closely related to the type and quality of diet that these chickens receive. In this research, villus length showed no statistical difference in the duodenum and ileum in all treatments compared to the control samples. However, the highest villus length was evidenced at the jejunum level with the 3% ECPF diet. Hosseini-Vashan et al. (2023) [59] reported similar villus length values in the jejunum when evaluating a feed with 0.6% coffee flour in broiler chickens.

Phenolic compounds play an important role in the development of intestinal villi [63]. Yatalaththov et al. (2021) [64] mention that the flavonoid content, tannins, and polyphenols present in coffee pulp improve the histology and morphometry of the jejunum in mice that are challenged with 10% ethanol, evidencing their regenerative action.

Hosseini-Vashan et al. (2023) [59], when evaluating the presence of 1.2% of green coffee pulp flour in the diet of broiler chickens, reported values similar to those found in this research in terms of villus length and crypt depth in the jejunum when 6% ECPF was used.

The crypt depth in chickens given feed with 0.9% of green coffee pulp had a similar response to that seen with doses of 1.5% compared to the control [59]. In this research, doses of 1.5% ECPF showed statistically superior results compared to the control samples in the jejunum and ileum.

The depth of intestinal crypts plays a crucial role in nutrient malabsorption and the regulation of gastrointestinal function. Previous research has indicated that deeper crypts can lead to increased secretion in the gastrointestinal tract and the reduced performance of the intestinal channel [62,65], due to the presence of toxins and pathogenic bacteria [66].

However, it is important to highlight that crypt depth can be influenced by different factors. For example, when crypt depth is related to the intake of soluble fiber, it can increase the number of enterocytes that are available for villus growth, which improves nutrient absorption [67]. Similar results were obtained in this study when 1.5% of ECPF was used. Coffee pulp is characterized by its high concentrations of pectin, a component that has been shown to increase the depth of intestinal crypts, as observed in studies conducted on rats [68]. Additionally, it has been suggested that coffee pulp possesses antibacterial properties [69,70], which could benefit intestinal health by inhibiting the growth of pathogenic bacteria.

The V/C level in the jejunum and ileum of the control treatment was higher compared to that with the use of 1.5% ECPF, showing lower values than those reported by the authors of [59] when evaluating the V/C in the jejunum of broiler chickens that were given a feed with 1.2% of green coffee flour. The V/C value is an indicator of the digestive capacity of the small intestine and it can be affected by both the quality and quantity of fiber in the diet and the age of the birds [65]. These results demonstrate that performance deteriorated as the levels of replacement increased. Studies on the effect of coffee on the gastrointestinal tract have been conducted more widely in humans but very little in animal models; hence, they yield controversial results. Iriondo-DeHond et al. (2021) [71] mention that some coffee components, such as caffeine and melanoidins, might increase intestinal motility, although the mechanisms are not clear. They also reported that coffee consumption showed positive regulation of the population of *Lactobacillus* spp. and *Bifidobacterium* spp. and a decrease in *Escherichia coli* and *Clostridium* spp. at the level of the proximal colon in specific pathogen-free A/J mice [72]. Similarly, the authors of [73] observed that coffee pulp supplementation in Wistar rats maintained the intestinal microbiota diversity.

An increase in intestinal surface area is closely linked to a significant improvement in digestive efficiency and nutrient absorption. This correlation is due to an increase in enzyme production at the brush border [74]. However, the development of the villus surface area can vary, depending on the age [75] and breed of the bird [76]. In this study, it was observed that extruded coffee pulp flour elicited a response similar to that in the control group in terms of intestinal surface results, suggesting a promising alternative for poultry feeding.

It has been shown that caffeine is rapidly absorbed at the stomach level in rats [77], while some polyphenols, such as chlorogenic acid (CGA), are absorbed by one-third in the small intestine. The rest of the polyphenols reach the colon, where they are degraded into simpler molecules by microbial action, as seen in subjects with an ileostomy [78]. To absorb coffee polyphenols, these must be hydrolyzed by the digestive enzymes, intestinal microbial population, or both [79].

Phenolic compounds must be broken down into small metabolites for proper absorption at the colon level [80]. Other factors, such as pH, temperature, bile salt concentration, and digestive enzyme activity also play a role [81].

Despite the existing literature offering limited information on the use of extruded coffee pulp in poultry feeding, the findings of this study establish a solid foundation for future research in this field. These results invite a more profound exploration of the topic and underscore the importance of efficiently utilizing agricultural waste, steering production toward circular economy principles.

## 5. Conclusions

This study demonstrated that substituting 25% of wheat bran with extruded coffee pulp flour in the diet of broiler chickens, representing 1.5% of the total diet in the finishing phase, showed similar productive parameters.

Additionally, the study showed that ECPF stimulates the epithelial turnover of villi in the jejunum and ileum, suggesting a potential benefit in terms of intestinal morphometry. However, it is crucial to highlight that this effect depends on the dosage used and this must be carefully balanced to avoid adverse impacts on productive performance.

Extruded coffee pulp flour is a potential ingredient to replace up to 1.5% of the diet in the feeding finishing phase in broiler chickens.

## Figures and Tables

**Figure 1 animals-14-01170-f001:**
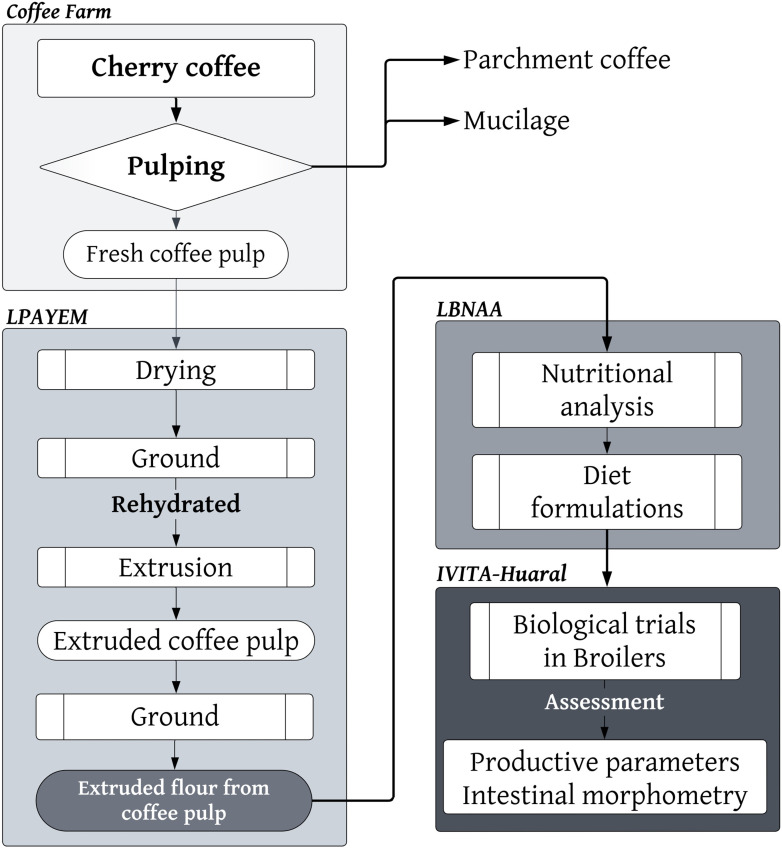
Methodological framework used in this study.

**Figure 2 animals-14-01170-f002:**
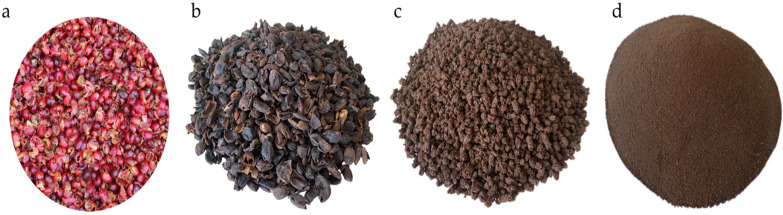
(**a**) Fresh coffee pulp, (**b**) dehydrated coffee pulp, (**c**) extruded coffee pulp, and (**d**) extruded coffee pulp flour (ECPF).

**Table 1 animals-14-01170-t001:** Composition of broiler finisher diets (35 to 41 days) containing different replacement levels of wheat bran (WB) with extruded coffee pulp flour (ECPF)).

Ingredients, %	Treatments				
	T1	T2	T3	T4	T5
Corn	56.62	55.31	54.01	52.70	51.40
Soybean meal	23.24	22.30	21.36	20.41	19.47
Flour whole soybean	6.69	8.71	10.73	12.75	14.77
Extruded coffee pulp flour	0.00	1.50	3.00	4.50	6.00
Wheat bran	6.00	4.50	3.00	1.50	0.00
Soybean oil	4.21	4.44	4.67	4.90	5.14
Diammonium phosphate	1.25	1.27	1.28	1.29	1.30
Calcium carbonate	0.78	0.76	0.75	0.74	0.72
NaCl	0.29	0.30	0.30	0.30	0.30
Methionine	0.24	0.25	0.25	0.25	0.26
Lysine-HCL	0.18	0.17	0.17	0.16	0.16
Vitamin and trace mineral premix	0.12	0.12	0.12	0.12	0.12
Sodium bicarbonate	0.10	0.10	0.10	0.10	0.10
Choline chloride	0.10	0.10	0.10	0.10	0.10
Mycotoxin sequestrant	0.10	0.10	0.10	0.10	0.10
Threonine	0.07	0.07	0.07	0.07	0.07
Total	100	100	100	100	100

**Table 2 animals-14-01170-t002:** Nutritional values of the diets.

Nutrient	Unit	Extruded Coffee Pulp Flour (Replacement Levels %)				
		T1:0	T2:25	T3:50	T4:75	T5:100
Metabolizable energy	KJ	12,866	12,866	12,866	12,866	12,866
Crude protein	%	18.68	18.85	19.02	19.18	19.35
Crude fiber	%	3.50	3.50	3.50	3.50	3.50
Calcium	%	0.73	0.73	0.73	0.73	0.73
Phosphorus available	%	0.37	0.37	0.37	0.37	0.37
Calcium/phosphorus		2.00	2.00	2.00	2.00	2.00
Sodium	%	0.16	0.16	0.16	0.16	0.16
Chlorine	%	0.27	0.27	0.26	0.26	0.26
Electrolyte balance	mEq/K	−5.80	−5.30	−4.90	−4.40	−4.00
Digestible lysine	%	1.01	1.01	1.01	1.01	1.01
Digestible methionine	%	0.51	0.52	0.52	0.52	0.52
Digestible methionine + cysteine	%	0.78	0.78	0.78	0.78	0.78
Digestible threonine	%	0.68	0.68	0.68	0.68	0.68
Digestible tryptophan	%	0.19	0.19	0.19	0.19	0.19
Digestible arginine	%	1.12	1.12	1.12	1.12	1.12
Digestible valine	%	0.78	0.78	0.78	0.78	0.78
Linoleic acid	%	4.28	4.54	4.80	5.07	5.33
Choline	ppm	520.74	520.74	520.74	520.74	520.74
Phytic phosphorus	%	0.28	0.27	0.25	0.24	0.23

**Table 3 animals-14-01170-t003:** Productive parameters of broilers from 35 to 41 days of age given feeds with 5 levels of ECPF as the replacement of a percentage of the WB in the finishing phase.

	Extruded Coffee Pulp Flour (Replacement Levels %)					ANOVA *p*-Value
	T1:0	T2:25	T3:50	T4:75	T5:100	
Feed Intake (g)	1320.00 ^a^	1320.00 ^a^	1290.00 ^a^	1209.00 ^b^	1151.00 ^c^	<0.001
Weight gain (g)	809.80 ^ab^	834.61 ^a^	784.26 ^ab^	761.13 ^b^	662.15 ^c^	<0.001
Feed conversion ratio	1.64 ^ab^	1.58 ^b^	1.65 ^ab^	1.59 ^b^	1.75 ^a^	0.0831

Different superscript letters (a, b, c) in a row indicate statistical differences (*p* < 0.05).

**Table 4 animals-14-01170-t004:** Morphometric measurements (in μm) of the small intestine segments (duodenum, jejunum, and ileum) in broilers at 41 days of age that were fed 5 levels of ECPF in place of WB in the finishing phase (35 to 41 days).

Parameter	Extruded Coffee Pulp Flour (Replacement Levels %)					ANOVA
	T1:0	T2:25	T3:50	T4:75	T5:100	*p*-Value
	Duodenum					
VL	1591.88 ^a^	1544.83 ^a^	1437.01 ^a^	1530.84 ^a^	1538.54 ^a^	0.55
VW	137.30 ^b^	136.22 ^b^	124.35 ^b^	136.92 ^b^	172.18 ^a^	0.0295
CD	173.77 ^c^	193.95 ^bc^	197.89 ^bc^	234.59 ^a^	227.18 ^ab^	< 0.001
Ratio V/C	9.41 ^a^	8.17 ^ab^	7.25 ^b^	6.68 ^b^	6.84 ^b^	< 0.001
VSA (μm^2^)	0.44 ^a^	0.40 ^a^	0.41 ^a^	0.38 ^a^	0.45 ^a^	0.19
	Jejunum					
VL	1380.64 ^a^	1321.42 ^ab^	1415.64 ^a^	1264.97 ^ab^	1191.04 ^b^	0.0279
VW	133.75 ^a^	161.79 ^a^	175.62 ^a^	145.88 ^a^	132.30 ^a^	0.699
CD	158.38 ^b^	199.51 ^a^	199.13 ^a^	218.34 ^a^	197.09 ^a^	0.0126
Ratio V/C	8.79 ^a^	6.85 ^b^	7.17 ^b^	5.90 ^b^	6.07 ^b^	< 0.001
VSA (μm^2^)	0.58 ^a^	0.67 ^a^	0.76 ^a^	0.58 ^a^	0.50 ^a^	0.21
	Ileum					
VL	952.67 ^a^	869.71 ^a^	932.79 ^a^	916.65 ^a^	895.83 ^a^	0.692
VW	146.77 ^a^	149.57 ^a^	142.25 ^a^	130.73 ^a^	158.88 ^a^	0.896
CD	144.21 ^c^	185.02 ^b^	186.69 ^b^	228.56 ^a^	212.09 ^ab^	< 0.001
Ratio V/C	6.89 ^a^	4.94 ^b^	5.06 ^b^	4.19 ^b^	4.24 ^b^	0.0028
VSA (μm^2^)	0.69 ^a^	0.67 ^a^	0.56 ^a^	0.66 ^a^	0.84 ^a^	0.92

Different letters in the rows indicate statistical differences (*p* < 0.05). VL: villus length, VW: villus width, CD: crypt depth, V/C: villus length/crypt depth ratio, VSA: villus surface area.

## Data Availability

The authors declare that the data are kept temporarily private because said information is part of a patent registration that is in progress.

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
