# Peer review of "Effect of Different Levels of Extruded Coffee (Coffea arabica) Pulp Flour on the Productive Performance and Intestinal Morphometry of Cobb 500 Broiler Chickens"

_animals, 2024, doi:10.3390/ani14081170_

Round 1

Reviewer 1 Report

Comments and Suggestions for Authors

Specify energy in KJ or MJ.
Table 1: Not salt but NaCl
Analyses of feed mixtures would be very interesting.  

Also information on the content of negative ingredients (caffeine, tannins, etc. ) in the coffee pulp would be interesting.

1. What is the main question of the study?
According to the title, the study aims to clarify the following questions: • Influence of the use of extruded coffee pulp on performance • Influence of the use of extruded coffee pulp on idestinal morphometry 2. 2. Do you consider the topic original or relevant in the field? Does it address a specific gap in the field?
• The description of the influence of coffee pulp on idestinal morphometry is relevant. 3. What does it add to the field compared to other published Material?
• Influence of the use of extruded coffee pulp on idestinal morphometry 4. What specific improvements should the authors consider regarding the methodology? What further controls should be considered?
• Analyses of compound feedingstuffs would be very interesting. Also information on the content of negative ingredients (caffeine, tannins, etc. ) in the coffee pulp
• The test in chickens in the last week of fattening does not allow any conclusion to be drawn on the possibility of use in feeding younger animals. 5. Are the conclusions consistent with the evidence and arguments presented? and do they address the main question posed?
• Conclusion:
o Indication of quantities used (1. 5%) instead of 25% substitute o No conclusions for idestinal morphometry available 7. Please include any additional comments on the tables and figures. • Table 2: o Label the columns analogous to Table 1 with T1, T2, etc. o Specify energy in KJ or MJ. o Specification of units for weight gain o Specification of feed intake without decimal places o Enter the P value with the same number of decimals. Use à < 0. 001 for very small values. Do not use ***. P-value already indicates the significance. - Table 3: o Label the columns analogous to Table 1 with T1, T2, etc. o Enter the P value with the same number of decimals. Use à < 0. 001 for very small values. Do not use *** P-value already indicates the significance. o Write the unit μm in the lines of the characteristics o For the characteristic V/C the unit μm does not apply! o Indication of values without decimal places!

Author Response

The reply to Reviewer 1 is attached.

Reviewer 2 Report

Comments and Suggestions for Authors

The manuscript entitled "Effect of different levels of extruded coffee (Coffea arabica) pulp flour on the productive performance and intestinal morphometry of Cobb 500 broiler chickens" was reviewed. Although the work is interesting, many doubts arise from the design, while the discussion is very difficult to follow. 

The authors must justify the rationale for including wheat bran in the control diet as it is a high-fiber, low-energy feed ingredient, while finishing broilers require a higher dietary concentration of ME, which can be achieved by adding more vegetable oil, but this increases the cost of the diet. Fiber from wheat bran plus fiber from whole soybean meal can result in lower feed digestibility.

It should also be described what was the basis for substituting wheat bran for coffee pulp. For example, if the substitution was made based on the ME or fiber contribution of both. 

For each of the five diets, the content of ME, limiting amino acids on a digestible basis, mainly lysine, methionine, threonine, and tryptophan, calcium, and available or non-phytic phosphorus, should be added.

The surface area of the villi must be calculated and shown in the tables. 

The discussion is extremely confusing. Numerical results found in other studies should be avoided as much as possible in order to compare them with those of the present investigation because they cut the fluidity of the reading.

In the Discussion, on lines 271-272 it is stated that "When crypts are deeper, it can

lead to nutrient malabsorption, increased gastrointestinal tract secretion, and lower carcass yield [60], [64]." However, with 25% inclusion of coffee pulp, CD was statistically higher in the jejunum and ileum than in the control, but these chicks gained the most weight. How can this be explained?

The surface area of the villi expresses the capacity of an organism to digest and assimilate the nutrients of a feed, and is more important than other measures of the villi, so it should be estimated and discussed.

Spell out acronyms when they first appear: (VA, TA, CP) in the paragraph between lines 265-273.

In the conclusions, it should be added that 25% of wheat bran can be substituted by coffee pulp when the total wheat bran in the diet is 6%, because if this percentage is different, then the conclusion is wrong. It can also be added that 1.5% of wheat bran can be substituted by coffee pulp. 25% coffee pulp did affect CD and V/C in jejunum and ileum. 

Comments on the Quality of English Language

Extensive editing of English language required

Author Response

The reply to Reviewer 2 is attached.

Round 2

Reviewer 2 Report

Comments and Suggestions for Authors

The article provides very good and clear information about the use of coffee pulp in broilers, so I consider it is ready to be published